# Weakly-Supervised Reinforcement Learning for Controllable Behavior

**Lisa Lee**[1,2]    **Benjamin Eysenbach**[1,2]    **Ruslan Salakhutdinov**[1]    **Shane Gu**[2]    **Chelsea Finn**[2,3]

[1]Carnegie Mellon University    [2]Google Brain    [3]Stanford University

`lslee@cs.cmu.edu`

## Abstract

Reinforcement learning (RL) is a powerful framework for learning to take actions to solve tasks. However, in many settings, an agent must winnow down the inconceivably large space of all possible tasks to the single task that it is currently being asked to solve. Can we instead constrain the space of tasks to those that are semantically meaningful? In this work, we introduce a framework for using weak supervision to automatically disentangle this semantically meaningful subspace of tasks from the enormous space of nonsensical "chaff" tasks. We show that this learned subspace enables efficient exploration and provides a representation that captures distance between states. On a variety of challenging, vision-based continuous control problems, our approach leads to substantial performance gains, particularly as the complexity of the environment grows.

## 1   Introduction

A general purpose agent must be able to efficiently learn a diverse array of tasks through interacting with the real world. The typical approach is to manually define a set of reward functions and only learn the tasks induced by these reward functions [18, 37]. However, defining and tuning the reward functions is labor intensive and places a significant burden on the user to specify reward functions for all tasks that they care about. Designing reward functions that provide enough learning signal yet still induce the correct behavior at convergence is challenging [34]. An alternative approach is to parametrize a family of tasks, such as goal-reaching tasks, and learn a policy for each task in this family [28, 38, 49, 61, 62, 71]. However, learning a single goal-conditioned policy for reaching all goals is a challenging optimization problem and is prone to underfitting, especially in high-dimensional tasks with limited data [12]. In this work, we aim to accelerate the acquisition of goal-conditioned policies by narrowing the goal space through weak supervision. Answering this question would allow an RL agent to prioritize exploring and learning meaningful tasks, resulting in faster acquisition of behaviors for solving human-specified tasks.

How might we constrain the space of tasks to those that are semantically meaningful? Reward functions and demonstrations are the predominant approaches to training RL agents, but they are expensive to acquire [34]. Generally, demonstrations require expert humans to be present [15, 20, 47], and it remains a challenge to acquire high-quality demonstration data from crowdsourcing [53]. In contrast, human preferences and ranking schemes provide an interface for sources of supervision that are easy and intuitive for humans to specify [11], and can scale with the collection of offline data via crowd-sourcing. However, if we are interested in learning many tasks rather than just one, these approaches do not effectively facilitate scalable learning of many different tasks or goals.

In this work, we demonstrate how weak supervision provides useful information to agents with minimal burden, and how agents can leverage that supervision when learning in an environment. We study one approach to using weak supervision in the goal-conditioned RL setting [2, 43, 57, 60, 66].

Instead of exploring and learning to reach every goal state, our weakly-supervised agent need only learn to reach states along meaningful axes of variation, ignoring state dimensions that are irrelevant to solving human-specified tasks. Critically, we propose to place such constraints through weak forms of supervision, instead of enumerating goals or tasks and their corresponding rewards. This weak

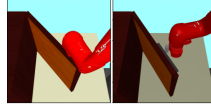

In which image...
...is the door opened wider?
...is the lighting brighter?
...is the robot closer to the door?

Figure 1: We propose weak supervision as a means to scalably introduce structure into goal-conditioned RL. The weak supervision is provided by answering binary questions based on the two images (left).

supervision is obtained by pairwise queries (see Figure 1), and our approach uses this supervision to learn a structured representation space of observations and goals, which can in turn be used to guide exploration, goal generation, and learning. Our approach enables the user to specify the factors of variation that matter for the efficient development of general-purpose agents.

The main contribution of this work is *weakly-supervised control (WSC)*, a simple framework for introducing weak supervision into RL. Our approach learns a semantically meaningful representation space with which the agent can generate its own goals, acquire distance functions, and perform directed exploration. WSC consists of two stages: we first learn a disentangled representation of states from weakly-labeled offline data, then we use the disentangled representation to constrain the exploration space for RL agents. We empirically show that learning disentangled representations can speed up reinforcement learning on various manipulation tasks, and improve the generalization abilities of the learned RL agents. We also demonstrate that WSC produces an *interpretable* latent policy, where latent goals directly align with controllable features of the environment.

## 2 Preliminaries

In this section, we overview notation and prior methods that we build upon in this work.

**Goal-conditioned RL**: We define a finite-horizon goal-conditioned Markov decision process by a tuple $(\mathcal{S}, \mathcal{A}, P, H, \mathcal{G})$ where $\mathcal{S}$ is the observation space, $\mathcal{A}$ is the action space, $P(s' \mid s, a)$ is an unknown dynamics function, $H$ is the maximum horizon, and $\mathcal{G} \subseteq \mathcal{S}$ is the goal space. In goal-conditioned RL, we train a policy $\pi_\theta(a_t \mid s_t, g)$ to reach goals from the goal space $g \sim \mathcal{G}$ by optimizing the expected cumulative reward $\mathbb{E}_{g \sim \mathcal{G}, \tau \sim (\pi, P)} \left[ \sum_{s \in \tau} R_g(s) \right]$, where $R_g(s)$ is a reward function defined by some distance metric between goals $g \in \mathcal{G}$ and observations $s \in \mathcal{S}$. In low-dimensional tasks, the reward can simply be defined as the negative $\ell_2$-distance in the state space [2]. However, defining distance metrics is more challenging in high-dimensional spaces such as images [81]. Prior work on visual goal-conditioned RL [57, 61] train an additional state representation model, such as a VAE encoder, and train a policy over encoded states and goals using $\ell_2$-distance in latent space as the reward. In this work, we accelerate the training of goal-conditioned RL by using a (learned) weakly-supervised disentangled representation to guide exploration and goal generation.

**Weakly-supervised disentangled representations**: Our approach leverages weakly-supervised disentangled representation learning in the context of reinforcement learning. Disentangled representation learning aims to learn interpretable representations of data, where each latent dimension measures a distinct *factor of variation*, conditioned on which the data was generated (see Fig. 2 for examples of factors). More formally, consider data-generating processes where $(f_1, \ldots, f_K) \in \mathcal{F}$ are the factors of variation, and observations $s \in \mathcal{S}$ are generated from a function $g^* : \mathcal{F} \to \mathcal{S}$. We would like to learn a disentangled latent representation $e : \mathcal{S} \to \mathcal{Z}$ such that, for any factor subindices $\mathcal{I} \subseteq [K]$, the subset of latent values $e_{\mathcal{I}}(s) = z_{\mathcal{I}}$ are only influenced by the true factors $f_{\mathcal{I}}$, and conversely, $e_{\backslash \mathcal{I}}(s) = z_{\backslash \mathcal{I}}$ are only influenced by $f_{\backslash \mathcal{I}}$.

We consider a form of weak supervision called *rank pairing*, where data samples consist of pairs of observations $\{s_1, s_2\}$ and weak binary labels $y \in \{0, 1\}^K$, where $y_k = \mathbf{1}(f_k(s_1) < f_k(s_2))$ indicates whether the $k$th factor value of observation $s_1$ is smaller than the corresponding factor value of $s_2$. Using this data, the weakly-supervised method proposed by Shu et al. [68] trains a discriminator $D$, a generator $G : \mathcal{Z} \to \mathcal{S}$, and an encoder $e : \mathcal{S} \to \mathcal{Z}$ to approximately invert $G$:

$$\min_D \quad \mathbb{E}_{(s_1, s_2, y) \sim \mathcal{D}} \left[ D(s_1, s_2, y) \right] + \mathbb{E}_{z_1, z_2 \sim N(0, I)} \left( 1 - D(G(z_1), G(z_2), y^{\text{fake}}) \right)$$

$$\max_G \quad \mathbb{E}_{z_1, z_2 \sim N(0, I)} \left[ D(G(z_1), G(z_2), y^{\text{fake}}) \right], \qquad \max_e \quad \mathbb{E}_{z \sim N(0, I)} \left[ e(z \mid G(z)) \right] \qquad (1)$$

where $y^{\text{fake}} = \mathbf{0}$ are fake labels. This approach is guaranteed to recover the true disentangled representation under mild assumptions [68]. We build upon their work in two respects. First, while

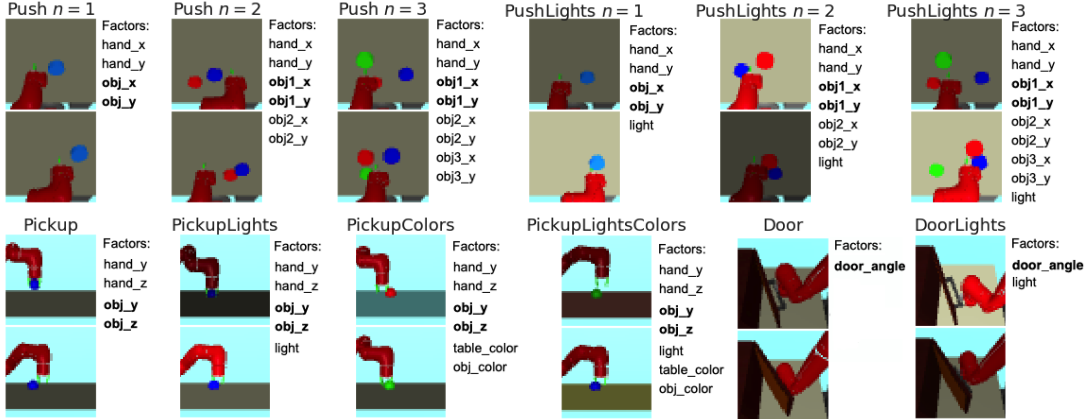

Figure 2: Our method uses weak supervision to direct exploration and accelerate learning on visual manipulation tasks. We extend the environments from [57] to make the tasks considerably harder by adding distractor objects (Push $n > 1$), randomized lighting (e.g., *PickupLights*), and randomized colors (e.g. *PickupColors*). Each data sample consists of a pair of image observations $\{s_1, s_2\}$ and factor labels $y_k = \mathbf{1}(f_k(s_1) < f_k(s_2))$ that indicate whether the $k$th factor of image $s_1$ has smaller value than that of image $s_2$. Example factors include the gripper position, object positions, brightness, and door angle. While these factors of variation could be directly measured with additional sensors in a controlled environment, weak supervision will be more useful in complex domains where such instrumentation is challenging. We only need to collect labels for the axes of variation that may be relevant for future downstream tasks (see Appendix A.2). Bolded factors correspond to the user-specified factor indices $\mathcal{I}$ indicating which of the factors are relevant for solving the class of tasks (see Sec. 3)

Shu et al. [68] used balanced and clean datasets, our method works with unbalanced classes, object occlusion, and varying lighting conditions. Second, we show how the learned representations can be used to accelerate RL.

## 3 The Weakly-Supervised RL Problem

Unlike standard RL, which requires hand-designed reward functions that are often expensive to obtain in complex environments, we aim to design the weakly-supervised RL problem in a way that provides a convenient form of supervision that scales with the collection of offline data. We aim to avoid labels in the loop of RL, precise segmentations, and numerical coordinates of objects.

Consider an environment with high complexity and large observation space such that it is intractable for an agent to explore the entire state space. Suppose that we have access to an offline dataset of weakly-labeled observations, where the labels capture semantically meaningful properties about the environment that are helpful to solving downstream tasks. How can a general-purpose RL agent leverage this dataset to learn new tasks faster? In this section, we formalize this problem statement.

**Problem Statement.** *Assume we are given a weakly-labelled dataset $\mathcal{D} := \{(s_1, s_2, y)\}$, which consists of pairs of observations $\{s_1, s_2\}$ and weak binary labels $y \in \{0, 1\}^K$, where $y_k = \mathbf{1}(f_k(s_1) < f_k(s_2))$ indicates whether the $k$-th factor value for observation $s_1$ is smaller than the corresponding factor value for $s_2$. Beyond these labels, the user also specifies a set of $K$ factors, and a subset of indices $\mathcal{I} \subseteq [K]$ specifying which of the factors are relevant for solving a class of tasks. During training, the agent may interact with the environment, but receives no supervision (e.g. no rewards) beyond the weak labels in $\mathcal{D}$. At test time, an unknown goal factor $f_{\mathcal{I}}^* \in \mathcal{F}_{\mathcal{I}}$ is sampled, and the agent receives a goal observation, e.g. a goal image, whose factors are equal to $f_{\mathcal{I}}^*$. The agent's objective is to learn a latent-conditioned RL policy that minimizes the goal distance: $\min_{\pi} \ \mathbb{E}_{\pi} \ d(f_{\mathcal{I}}(s), f_{\mathcal{I}}^*)$.*

Weakly-supervised RL may be useful in many real-world scenarios, as weak-supervision is often cheaper to collect than expert demonstrations. For a visual manipulation task (Fig. 2), labels might indicate the relative position of the robot gripper arm between two image observations; the corresponding goal factor space $\mathcal{F}_{\mathcal{I}}$ might be the XY-position of an object. Note that we only need to collect labels for the axes of variation that may be relevant for future downstream tasks (see Appendix A.2). At test time, the agent receives a goal image observation, and is evaluated on how closely it can move the object to the goal location.

The next section develops a framework for solving the weakly-supervised RL problem. Then Sec. 5 empirically investigates whether weak supervision accelerates RL in an economical way.

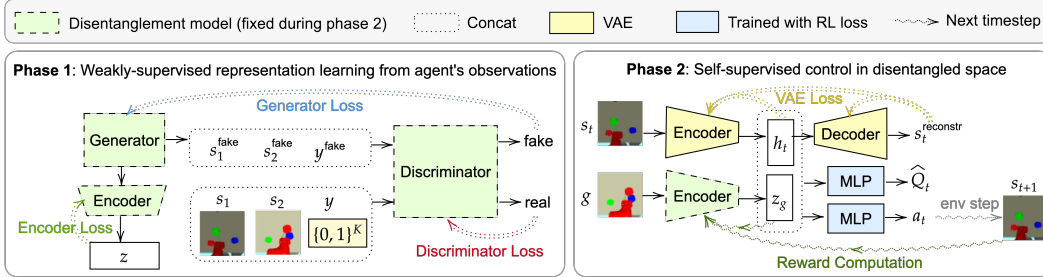

**Figure 3: Weakly-Supervised Control framework**. *Left*: In Phase 1, we use the weakly-labelled dataset to learn a disentangled representation by optimizing the losses in Eq. 1. *Right*: In Phase 2, we use the learned disentangled representation to guide goal generation and define distances. At the start of each episode, the agent samples a latent goal $z_g$ either by encoding a goal image $g$ sampled from the replay buffer, or by sampling directly from the latent goal distribution (Eq. 2). The agent samples actions using the goal-conditioned policy, and defines rewards as the negative $\ell_2$ distance between goals and states in the disentangled latent space (Eq. 3).

## 4 Weakly-Supervised Control

In this section, we describe a simple training framework for the weakly-supervised RL problem. Our *weakly-supervised control (WSC)* framework consists of two stages: we first learn a disentangled representation from weakly-labelled RL observations, and then use this disentangled space to guide the exploration of goal-conditioned RL along semantically meaningful directions.

### 4.1 Learning disentangled representations from observations

We build upon the work of Shu et al. [68] for learning disentangled representations, though other methods could be used. Their method trains an encoder, generator, and discriminator by optimizing the losses in Eq. 1. After training the disentanglement model, we discard the discriminator and the generator, and use the encoder to define the goal space and compute distances between states.

While Shu et al. [68] assumes that all combinations of factors are present in the dataset and that data classes are perfectly balanced (i.e., exactly one image for every possible combination of factors), these assumptions usually do not hold for significantly less clean data coming from an agent's observations in a physical world. For example, not all factor combinations are physically possible to achieve: an object cannot be floating in mid-air without a robot gripper holding it, and two solid objects cannot occupy the same space at the same time. This affects the data distribution: for example, when the robot is holding the object in *Pickup*, there is high correlation between the gripper and object positions. Another issue is partial observability: the agent may lack sensors to observe some aspects of its environment, such as being unable to see through occlusions.

To generate the Sawyer datasets shown in Fig. 2, we corrected the sampled factor combinations to be physically feasible before generating the corresponding image observations in the Mujoco simulator. Furthermore, to reflect the difficulty of collecting a large amount of samples in complex RL environments, we only sampled 256 or 512 images in the training dataset, which is much smaller than the combinatorial size of toy datasets such as dSprites [54] (737,280 images).

Empirically, we found that it is more challenging to learn a disentangled representation on the Sawyer observations (see Table 3), yet we show in Sec. 5 that imperfect disentanglement models can still drastically accelerate training of goal-conditioned RL. In the next section, we describe how we use the learned disentangled space to generate goals, define reward functions, and do directed exploration.

### 4.2 Structured Goal Generation & Distance Function

Next, we describe how our method uses the learned disentangled model $e : \mathcal{S} \rightarrow \mathcal{Z}$ and the user-specified factor indices $\mathcal{I} \subseteq [K]$ to train a goal-conditioned policy. The agent will propose its own goals to practice, attempt the proposed goals, and use the experience to update its policy.

Our method defines the goal space to be the learned disentangled latent space $\mathcal{Z}_{\mathcal{I}}$, restricted to the indices in $\mathcal{I}$. The goal sampling distribution is defined as

$$p(\mathcal{Z}_{\mathcal{I}}) := \mathrm{Uniform}(\mathcal{Z}_{\mathcal{I}}^{\min}, \mathcal{Z}_{\mathcal{I}}^{\max}), \tag{2}$$

where $\mathcal{Z}_{\mathcal{I}}^{\min} = \min_{s \in \mathcal{D}} e_{\mathcal{I}}(s)$ and $\mathcal{Z}_{\mathcal{I}}^{\max} = \max_{s \in \mathcal{D}} e_{\mathcal{I}}(s)$ denote the elementwise min and max over the dataset, and $\mathrm{Uniform}(\cdot, \cdot)$ denotes the uniform distribution.

In each iteration, our method samples latent goals $z_g \in \mathcal{Z}_{\mathcal{I}}$ by either sampling from $p(\mathcal{Z}_{\mathcal{I}})$, or sampling an image observation from the replay buffer and encoding it with the disentangled model, $z_g = e_{\mathcal{I}}(s_g)$. Then, our method attempts this goal by executing the policy to get a trajectory $(s_1, a_1, ..., s_T)$. When sampling transitions $(s_t, a_t, s_{t+1}, z_g)$ from the replay buffer for RL training, we use hindsight relabeling [2] with corrected goals to provide additional training signal. In other words, we sometimes relabel the transition $(s_t, a_t, s_{t+1}, z'_g)$ with a corrected goal $z'_g$, which is sampled from either the goal distribution $p(\mathcal{Z}_{\mathcal{I}})$ in Eq. 2, or from a future state in the current trajectory. Our method defines the reward function as the negative $\ell_2$-distance in the disentangled latent space:

$$r_t := R_{z_g}(s_{t+1}) := -\|e_{\mathcal{I}}(s_{t+1}) - z_g\|_2^2. \quad (3)$$

We summarize our weakly-supervised control (WSC) framework in Fig. 3 and Alg. 1. We start by learning the disentanglement module using the weakly-labelled data. Next, we train the policy with off-policy RL, sampling transitions with hindsight relabeling. At termination, our method outputs a goal-conditioned policy which is trained to go to a state that is close to the goal in the disentangled latent space.

---

**Algorithm 1** Weakly-Supervised Control

**Input**: Weakly-labeled data $\mathcal{D}$, factor indices $\mathcal{I} \subseteq [K]$
  Train disentangled representation $e(s)$ using $\mathcal{D}$.
  Compute $\mathcal{Z}_{\mathcal{I}}^{\min} = \min_{s \in \mathcal{D}} e_{\mathcal{I}}(s)$.
  Compute $\mathcal{Z}_{\mathcal{I}}^{\max} = \max_{s \in \mathcal{D}} e_{\mathcal{I}}(s)$.
  Define $p(\mathcal{Z}_{\mathcal{I}}) := \mathrm{Uniform}(\mathcal{Z}_{\mathcal{I}}^{\min}, \mathcal{Z}_{\mathcal{I}}^{\max})$.
  Initialize replay buffer $\mathcal{R} \leftarrow \emptyset$.
  **for** iteration= 0, 1, . . . , **do**
    Sample a goal $z_g \in \mathcal{Z}$ and an initial state $s_0$.
    **for** $t = 0, 1, \ldots, H-1$ **do**
      Get action $a_t \sim \pi(s_t, z_g)$.
      Execute action and observe $s_{t+1} \sim p(\cdot \mid s_t, a_t)$.
      Store $(s_t, a_t, s_{t+1}, z_g)$ into replay buffer $\mathcal{R}$.
    **for** $t = 0, 1, \ldots, H-1$ **do**
      **for** $j = 0, 1, \ldots, J$ **do**
        With probability $p$, sample $z'_g \sim p(\mathcal{Z}_{\mathcal{I}})$. Otherwise, sample a future state $s' \in \tau_{>t}$ in the current trajectory and compute $z'_g = e_{\mathcal{I}}(s')$.
        Store $(s_t, a_t, s_{t+1}, z'_g)$ into $\mathcal{R}$.
    **for** $k = 0, 1, \ldots, N-1$ **do**
      Sample $(s, a, s', z_g) \sim \mathcal{R}$.
      Compute $r = R_{z_g}(s') = -\|e_{\mathcal{I}}(s') - z_g\|_2^2$.
      Update actor and critic using $(s, a, s', z_g, r)$.
  **return** $\pi(a \mid s, z)$

---

## 5 Experiments

We aim to first and foremost answer our core hypothesis: (1) Does weakly-supervised control help guide exploration and learning, for increased performance over prior approaches? Further we also investigate: (2) What is the relative importance of disentanglement for goal generation versus for distances?, (3) Is the policy's behavior interpretable?, (4) Is weak supervision necessary for learning a disentangled state representation?, (5) How much weak supervision is needed to learn a sufficiently disentangled state representation, and (6) How much error can be tolerated in the labeling? Questions 1∼3 are investigated in this section, while questions 4∼6 are studied in Appendix A.

To answer these questions, we consider several vision-based, goal-conditioned manipulation tasks shown in Fig. 2. We extend the environments from [57] to make the tasks considerably harder, by adding distractor objects, randomized lighting, and randomized colors. In environments with 'light' as a factor (e.g., *PushLights*), the lighting conditions change randomly at the start of each episode. In environments with 'color' as a factor (e.g., *PickupColors*), both the object color and table color randomly change at the start of each episode (5 table colors, 3 object colors). In the *Push* and *Pickup* environments, the agent's task is to move a specific object to a goal location. In the *Door* environments, the agent's task is to open the door to match a goal angle. Both the state and goal observations are $48 \times 48$ RGB images.

**Comparisons**: We compare our method to prior state-of-the-art goal-conditioned RL methods, which are summarized in Table 1. While the original hindsight experience replay (HER) algorithm [2] requires the state space to be disentangled, this assumption does not hold in our problem setting, where the observations are high-dimensional images. Thus, in our experiments, we modified **HER** [2] to sample relabeled goals from the VAE prior $g \sim \mathcal{N}(0, I)$ and use the negative $\ell_2$-distance between goals and VAE-encoded states as the reward function. **RIG** [57] and **SkewFit** [61] are extensions of HER that use a modified goal sampling distribution that places higher weight on rarer states. RIG uses MLE to train the VAE, while SkewFit uses data samples from

| Method | $p(\mathcal{Z})$ | $R_{z_g}(s')$ |
|---|---|---|
| RIG | $\mathcal{N}(0, I)$ | $-\|e^{\mathrm{VAE}}(s') - z_g\|_2^2$ |
| SkewFit | $p^{\mathrm{skew}}(\mathcal{R})$ | $-\|e^{\mathrm{VAE}}(s') - z_g\|_2^2$ |
| WSC | $\mathrm{Unif}(\mathcal{Z}_{\mathcal{I}}^{\min}, \mathcal{Z}_{\mathcal{I}}^{\max})$ | $-\|e_{\mathcal{I}}(s') - z_g\|_2^2$ |

Table 1: Conceptual comparison between our method weakly-supervised control (WSC), and prior visual goal-conditioned RL methods, with their respective latent goal distributions $p(\mathcal{Z})$ and goal-conditioned reward functions $R_{z_g}(s')$. Our method can be seen as an extension of prior work to the weakly-supervised setting.

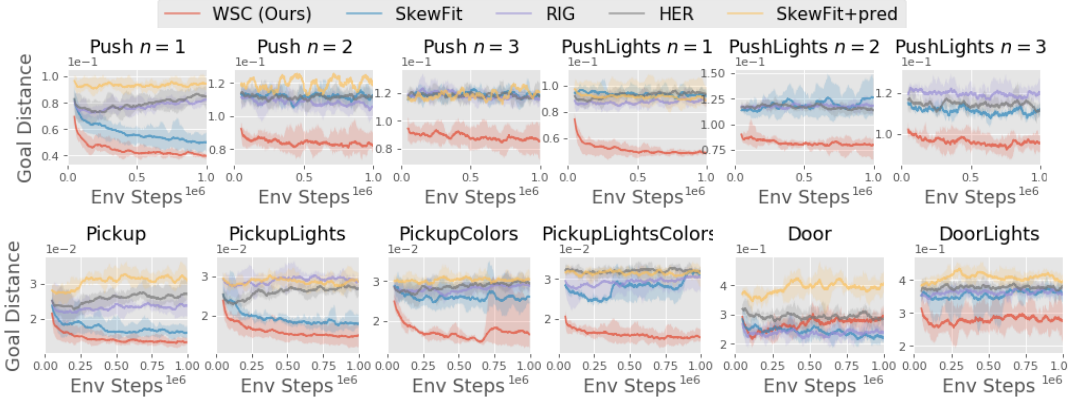

Figure 4: **Performance vs. training steps on visual goal-conditioned tasks**. Weakly-supervised control (WSC) learns more quickly than prior state-of-the-art goal-conditioned RL methods (HER, RIG, SkewFit), particularly as the complexity of the environment grows. Thus, we see that doing directed exploration and goal sampling in a (learned) semantically-disentangled latent space can be more effective than doing purely unsupervised exploration in the VAE latent space.

$p^{\text{skew}}(\mathcal{R})$ to train the VAE. For direct comparison, we use the weakly-labeled dataset $\mathcal{D}$ in HER, RIG, and SkewFit to pre-train the VAE, from which goals are sampled. Note that all algorithms used in our experiments have the same time complexity. See Appendix B for further implementation details.

Additionally, to investigate whether our disentanglement approach for utilizing weak suppervision is better than alternative methods, we compare to a variant of SkewFit that optimizes an auxiliary prediction loss on the factor labels, which we refer to as **Skewfit+pred**.

**Dataset generation**: Both the training and test datasets were generated from the same distribution, and each set consists of 256 or 512 images (see Table 6). To generate the Sawyer datasets shown in Fig. 2, we first sampled each factor value uniformly within their respective range, then corrected the factors to be physically feasible before generating the corresponding image observations in the Mujoco simulator. In *Push* environments with $n > 1$ objects, the object positions were corrected to avoid collision. In *Pickup* environments, we sampled the object position on the ground (obj_z=0) with 0.8 probability, and otherwise placed the object in the robot gripper (obj_z$\geq$ 0). In *Door* environments, the gripper position was corrected to avoid collision with the door.

**Eval metric**: At test-time, all RL methods only have access to the test goal image, and is evaluated on the true goal distance. In *Push* and *Pickup*, the true goal distance is defined as the $\ell_2$-distance between the current object position and the goal position. In *Push* environments with $n > 1$ objects, we only consider the goal distance for the blue object, and ignore the red and green objects (which are distractor objects to make the task more difficult). In *Door* environments, the true goal distance is defined as the distance between the current door angle and the goal angle value.

### 5.1 Does weakly-supervised control help guide exploration and learning?

Do the disentangled representations acquired by our method guide goal-conditioned policies to explore in more semantically meaningful ways? In Fig. 4, we compare our method to prior state-of-the-art goal-conditioned RL methods on visual goal-conditioned tasks in the Sawyer environments (see Fig. 2). We see that doing directed exploration and goal sampling in a (learned) disentangled latent space is substantially more effective than doing purely unsupervised exploration in VAE latent state space, particularly for environments with increased variety in lighting and appearance.

Then, a natural question remains: is our disentanglement approach for utilizing weak supervision better than alternative methods? One simple approach is to add an auxiliary loss to predict the factor identity from the representation. We train a variant of SkewFit where the final hidden layer of the VAE is also trained to optimize an auxiliary prediction loss on the factor identity, which we refer to as 'Skewfit+pred'. In Fig. 4, we find that Skewfit+pred performs worse than WSC even though it uses stronger supervision (exact labels).This comparison suggests that disentangling meaningful and irrelevant factors of the environment is important for effectively leveraging weak supervision.

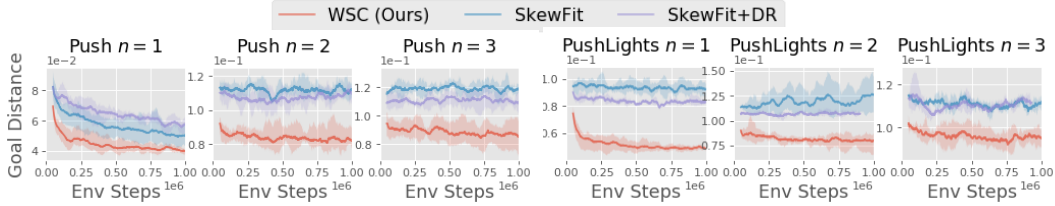

Figure 5: SkewFit+DR is a variant that samples goals in VAE latent space, but uses reward distances in disentangled latent space. We see that the disentangled distance metric can help slightly in harder environments (e.g., *Push* $n = 3$), but the goal generation mechanism of WSC is crucial for efficient exploration.

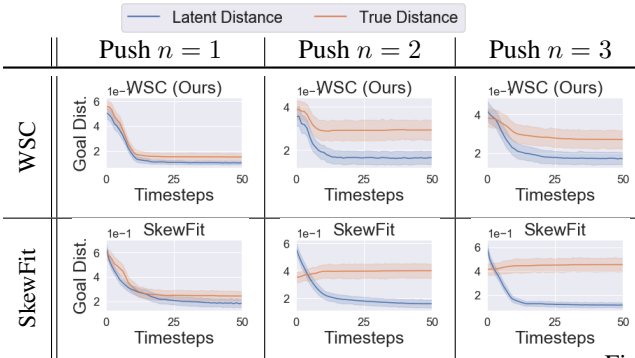

Figure 6: We roll out policies on the *Push* tasks after training. The disentangled distance optimized by WSC (top) is more indicative of the true goal distance than the latent distance optimized by SkewFit (bottom), especially for more complex tasks ($n > 1$).

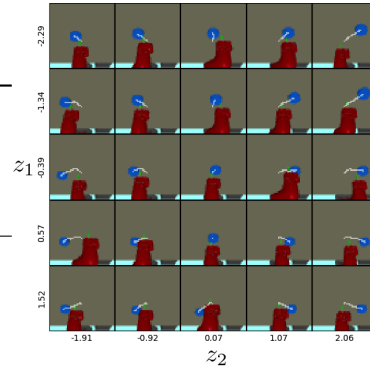

Figure 7: **Interpretable control**: We roll out a trained WSC policy conditioned on varying disentangled latent goals $(z_1, z_2) \in [\mathcal{Z}_\mathcal{I}^{\min}, \mathcal{Z}_\mathcal{I}^{\max}]$, and plot the trajectory of the object (white line). The latent goals directly align with the direction in which the policy moves the blue object.

## 5.2 Ablation: What is the role of distances vs. goals?

Our method uses the representation in two places: for goal-generation (Eq. 2) and for the distance metric (Eq. 3). Our next experiment studies the relative importance of using a disentangled representation in both places. First, we investigate whether the distance metric in the disentangled space alone is enough to learn goal-conditioned tasks quickly. To do so, we trained a variant of SkewFit that samples latent goals in VAE latent space, but uses distances in disentangled latent space as the reward function. In Fig. 5, we see that the disentangled distance metric can help slightly in harder environments, but underperforms compared to the full method (WSC) with goal generation in disentangled latent space. Thus, we conclude that both the goal generation mechanism and distance metric of our method are crucial components for enabling efficient exploration.

Next, we tested whether the distance metric defined over the learned disentangled representation provides a more accurate signal for the true goal distance. In Fig. 6, we evaluate trained policies on the *Push* tasks, and compare the latent goal distance vs. the true goal distance at every timestep. The disentangled distance optimized by WSC is more indicative of the true $\ell_2$ goal distance than the latent distance optimized by SkewFit, especially for more complex tasks ($n > 1$), suggesting that the disentangled representation provide a more accurate reward signal for the training agent.

## 5.3 Is the policy's latent space interpretable?

Since our method uses an interpretable latent goal space to generate self-proposed goals and compute rewards for training the policy, we checked whether the learned policy is also semantically meaningful. Table 2 in the Appendix measures the correlation between latent goals and the final states of the policy rollout. For various latent goals $z_g \in \mathcal{Z}$, we rolled out the trained policy $\pi(a \mid s, z_g)$ and compared the final state with the latent goal $z_g$ that the policy was conditioned on (see Section 5.3). For our method, we did a grid sweep over the latent goal values in $[\mathcal{Z}_\mathcal{I}^{\min}, \mathcal{Z}_\mathcal{I}^{\max}]$. For SkewFit, we took the latent dimensions that have the highest correlations with the true object XY positions, then did a similar grid sweep over the latent goal space. Our method achieves higher Pearson correlation between latent goals and final states, meaning that it learns a more interpretable goal-conditioned policy where the latent goals align directly with the final state of the trajectory rollout.

In Fig. 7, we visualize the trajectories generated by our method's policy when conditioned on different latent goals, obtained by doing a grid sweep over the latent space $[\mathcal{Z}_{\mathcal{I}}^{\min}, \mathcal{Z}_{\mathcal{I}}^{\max}]$. We fixed the initial object and gripper positions for each trajectory. We see that the latent goal values $z_g$ directly align with the final object position after rolling out the policy $\pi(a \mid s, z_g)$. In other words, varying each latent goal dimension corresponds to directly changing the object position in the X- or Y-coordinate.

## 6 Related Work

Reinforcement learning of complex behaviors in rich environments remains an open problem. Many of the successful applications of RL in prior work [6, 32, 69, 76] effectively operate in a regime where the amount of data (i.e., interactions with the environment) dwarfs the complexity of task at hand. Insofar as alternative forms of supervision is the key to success for RL methods, prior work has proposed a number of techniques for making use of various types of ancillary supervision.

A number of prior works incorporate additional supervision beyond rewards to accelerate RL. One common theme is to use the task dynamics itself as supervision, using either forward dynamics [19, 35, 44, 77, 82], some function of forward dynamics [14], or inverse dynamics [1, 58, 59] as a source of labels. Another approach explicitly predicts auxiliary labels [13, 30, 41, 67]. Compact state representations can also allow for faster learning and planning, and prior work has proposed a number of tools for learning these representations [4, 21, 27, 48, 51, 52, 57, 80]. Bengio et al. [5], Thomas et al. [72] propose learning representations using an independent controllability metric, but the joint RL and representation learning scheme has proven difficult to scale in environment complexity. Perhaps most related to our method is prior work that directly learns a compact representation of goals [31, 56, 61]. Our work likewise learns a low-dimensional representation of goals, but crucially learns it in such a way that we "bake in" a bias towards meaningful goals, thereby avoiding the problem of accidentally discarding salient state dimensions.

Human supervision is an important but expensive aspect of reward design [34], and prior work has studied how reward functions might be efficiently elicited from weak supervision. In settings where a human operator can manually control the system, a reward function can be acquired by applying inverse RL on top of human demonstrations [7, 20, 24, 28, 63]. Another technique for sample-efficient reward design is to define rewards in terms of pre-trained classifiers [25, 70, 74, 78], which might be learned with supervised learning. State marginal distributions, which can be easier to specify in some tasks, have also been used as supervision for RL [28, 49]. Our method utilizes a much weaker form of supervision than state marginals, which potentially allows it to scale to more complex tasks. A final source of supervisory signal comes in the form of human preferences or rankings [7, 11, 79], where humans provide weak supervision about which of two behaviors they prefer. Our approach similarly obtains weak supervision from humans, but uses it to acquire a disentangled space for defining many tasks, rather than directly defining a single task reward.

Many goal-conditioned RL algorithms learn a distance metric between states, using this learned distance for proposing goals [23, 60, 64, 75], estimating Q-values [22], planning [17, 65], and reward shaping [36]. A common strategy is to learn a latent representation of states such that the distance in latent space corresponds to a "meaningful" distance in state space [57, 61]. Our method employs this representation-based distance learning strategy for proposing goals and constructing reward functions. We accelerate this distance learning by making use of auxiliary weak supervision.

Finally, our approach leverages weakly-supervised disentangled representation learning in the context of RL. Learning such semantically-meaningful representations are useful for many downstream tasks that require learned models to be human-controllable or interpretable [29, 46, 73]. While there is no canonical definition for disentanglement, several formal definitions have been proposed [39, 68]. Many unsupervised methods for disentangled representation learning [9, 10, 16, 40, 45] attempt to approximate $g^*(\mathcal{F})$. However, unsupervised methods are generally brittle to hyperparameter settings and do not lead to consistently disentangled latent representations [50]. Recently, weakly-supervised disentanglement methods [8, 26, 68] have been shown to produce more robustly disentangled representations than unsupervised methods, without requiring large amounts of supervision.

## 7 Conclusion

We proposed weak supervision as a means to scalably introduce structure into goal-conditioned RL. To leverage the weak supervision, we proposed a simple two phase approach that learns a disentangled representation, and then uses it to guide exploration, propose goals, and inform a distance metric.

Our experimental results indicate that our approach, WSC, substantially outperforms self-supervised methods that cannot cope with the breadth of the environments. Further, our comparisons suggest that our disentanglement-based approach is critical for effectively leveraging the weak supervision.

Despite its strong performance, WSC has multiple limitations. While WSC has the ability to leverage weak labels that can be easily collected offline with approaches like crowd compute, WSC requires a user to indicate the factors of variation that are relevant for downstream tasks, which may require expertise. However, we expect the indication of such factors to still require substantially less expert effort than demonstrations or reward specification. Further, our method only uses weak supervision during pre-training, which may produce representations that do not always generalize to new interaction later encountered by the agent. Incorporating weak supervision online, in the loop of RL, could address this issue to improve performance. In such settings, we expect class imbalance and human-in-the-loop learning to present important, but surmountable challenges.

Looking forward, our results suggest a number of interesting directions for future work. For example, there may be other forms of weak supervision [68] that can provide useful signal to the agent, as well as other ways to leverage these labels. Given the promising results in increasingly complex environments, evaluating this approach with robots in real-world environments is an exciting future direction. Overall, we believe that our framework provides a new perspective on supervising the development of general-purpose agents acting in complex environments.

## Acknowledgements

We thank Yining Chen, Vitchyr Pong, Ben Poole, and Archit Sharma for helpful discussions and comments. We thank Michael Ahn for help with running the manipulation experiments, and also thank Benjamin Narin, Rafael Rafailov, and Riley DeHaan for help with initial experiments. LL is supported by the National Science Foundation (DGE-1745016). BE is supported by the Fannie and John Hertz Foundation and the National Science Foundation (DGE-1745016). RS is supported by NSF IIS1763562, ONR Grant N000141812861, and US Army. CF is a CIFAR Fellow. Any opinions, findings and conclusions or recommendations expressed in this material are those of the author(s) and do not necessarily reflect the views of the National Science Foundation.

## Broader Impact

We highlight two potential impacts for this work. Most immediately, weak supervision from humans may be an inexpensive yet effective step towards human-AI alignment [34, 55]. While prior work [3, 11, 42] has already shown how weak supervision in the form of *preferences* can be used to train agents, our work explores how a different type of supervision – invariance to certain factors – can be elicited from humans and injected as an inductive bias in an RL agent. One important yet delicate form of invariance is fairness. In many scenarios, we may want our agent to treat humans of different ages or races equally. While this fairness might be encoded in the reward function, our method presents an alternative, where fairness is encoded as observations' invariance to certain protected attributes (e.g., race, gender).

One risk with this work is misspecification of the factors of variation. If some factors are ignored, then the agent may require longer to solve certain tasks. More problematic is if spurious factors of variation are added to the dataset. In this case, the agent may be "blinded" to parts of the world, and performance may suffer. A question for future work is the automatic discovery of these spurious weak labels.

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
