[Supplementary Material · appendix.pdf]

| | | Pearson correlation | |
| Env | Factor | SkewFit | WSC (Ours) |
| --- | --- | --- | --- |
| Push $n = 1$ | obj_x | $0.94 \pm 0.03$ | $0.95 \pm 0.03$ |
| | obj_y | $0.66 \pm 0.17$ | $0.94 \pm 0.04$ |
| Push $n = 2$ | obj1_x | $0.59 \pm 0.50$ | $0.69 \pm 0.37$ |
| | obj1_y | $0.44 \pm 0.68$ | $0.86 \pm 0.05$ |
| Push $n = 3$ | obj1_x | $0.44 \pm 1.05$ | $0.78 \pm 0.11$ |
| | obj1_y | $0.38 \pm 1.44$ | $0.89 \pm 0.01$ |

Table 2: **Is the learned policy interpretable?** We measure the correlation between the true factor value of the final state in the trajectory vs. the corresponding latent dimension of the latent goal $z_g$ used to condition the policy. We show 95% confidence over 5 seeds. Our method attains higher correlation between latent goals and final states, meaning that it learns a more interpretable goal-conditioned policy.

| | | Pearson correlation | |
| Env | Factor | VAE (SkewFit) | WSC (Ours) |
| --- | --- | --- | --- |
| Push $n = 1$ | hand_x | $0.97 \pm 0.04$ | $0.97 \pm 0.01$ |
| | hand_y | $0.85 \pm 0.07$ | $0.93 \pm 0.02$ |
| | obj_x | $0.78 \pm 0.28$ | $0.97 \pm 0.01$ |
| | obj_y | $0.65 \pm 0.31$ | $0.95 \pm 0.01$ |
| Push $n = 3$ | hand_x | $0.95 \pm 0.03$ | $0.98 \pm 0.01$ |
| | hand_y | $0.50 \pm 0.33$ | $0.94 \pm 0.03$ |
| | obj1_x | $0.12 \pm 0.18$ | $0.96 \pm 0.01$ |
| | obj1_y | $0.15 \pm 0.03$ | $0.92 \pm 0.02$ |

Table 3: **Is the learned state representation disentangled?** We measure the correlation between the true factor value of the input image vs. the latent dimension of the encoded image on the evaluation dataset (95% confidence over 5 seeds). We find that unsupervised VAEs are often insufficient for learning a disentangled representation.

## A  Additional Experimental Results

### A.1  Is the learned state representation disentangled?

To see whether weak supervision is necessary to learn state representations that are disentangled, we measure the correlation between true factor values and the latent dimensions of the encoded image in Table. 3. For the VAE, we took the latent dimension that has the highest correlation with the true factor value. The results illustrate that unsupervised losses are often insufficient for learning a disentangled representation, and utilizing weak labels in the training process can greatly improve disentanglement, especially as the environment complexity increases.

### A.2  How much weak supervision is needed?

Our method relies on learning a disentangled representation from weakly-labelled data, $\mathcal{D} = \{(s_1^{(i)}, s_2^{(i)}, y^{(i)})\}_{i=1}^{N}$. However, the total possible number of pairwise labels for each factor of variation is $N = \binom{M}{2}$, where $M \in \{256, 512\}$ is the number of images in the dataset. In this section, we investigate how much weak supervision is needed to learn a sufficiently-disentangled state representation such that it helps supervise goal-conditioned RL.

**Number of factors that are labelled**: There can be many axes of variation in an image observation, especially as the complexity of the environment grows. For example, the *PushLights* environment with $n = 3$ objects has nine factors of variation, including the positions of the robot arm and objects, and lighting (see Figure 2).

In Figure 8, we investigate whether WSC requires weak labels for all or some of the factors of variation. To do so, we compared the performance of WSC as we vary the set of factors of variation that are weakly-labelled in the dataset $\mathcal{D}$. We see that WSC performs well even when weak labels are not provided for task-irrelevant factors of variation, such as hand position and lighting.

**Number of weak labels**: In Table 4, we evaluate the quality of the learned disentangled representation model as we vary the number of weak labels, $N$. We measure disentanglement by evaluating the Pearson correlation between the true factor value compared to the latent dimension. We observe that, even with only 1024 pairwise labels, the resulting representation has a good degree of disenganglement, i.e. Pearson correlation of 0.8 or higher.

In Figure 9, we evaluate the downstream performance of our method on visual goal-conditioned tasks as we vary the number of weak labels. We see that our method outperforms SkewFit when provided at least 1024, 1024, 256, and 128 weak labels for *Push* $n = 1$, *PushLights* $n = 3$, *PickupLightsColors*, and *DoorLights*, respectively. Further, we find that 1024 pairwise labels is generally sufficient for good performance on all domains.

Figure 8: **How many factors of variation need to be labelled?** WSC outperforms SkewFit even without being provided weak labels for task-irrelevant factors, such as hand position and lighting.

| | | | | | PushLights $n = 3$ | | | | |
| $N$ | hand_x | hand_y | obj1_x | obj1_y | obj2_x | obj2_y | obj3_x | obj3_y | light |
|---|---|---|---|---|---|---|---|---|---|
| 128 | $0.79 \pm 0.04$ | $0.64 \pm 0.05$ | $0.44 \pm 0.08$ | $0.32 \pm 0.05$ | $0.60 \pm 0.03$ | $0.51 \pm 0.05$ | $0.49 \pm 0.07$ | $0.41 \pm 0.06$ | $0.86 \pm 0.04$ |
| 256 | $0.87 \pm 0.02$ | $0.75 \pm 0.05$ | $0.58 \pm 0.04$ | $0.57 \pm 0.04$ | $0.60 \pm 0.04$ | $0.66 \pm 0.03$ | $0.65 \pm 0.07$ | $0.50 \pm 0.06$ | $0.90 \pm 0.02$ |
| 512 | $0.93 \pm 0.01$ | $0.86 \pm 0.01$ | $0.71 \pm 0.03$ | $0.70 \pm 0.05$ | $0.70 \pm 0.04$ | $0.58 \pm 0.05$ | $0.76 \pm 0.04$ | $0.67 \pm 0.05$ | $0.85 \pm 0.04$ |
| 1024 | $0.97 \pm 0.01$ | $0.91 \pm 0.01$ | $0.86 \pm 0.01$ | $0.81 \pm 0.02$ | $0.83 \pm 0.02$ | $0.80 \pm 0.03$ | $0.83 \pm 0.03$ | $0.80 \pm 0.02$ | $0.94 \pm 0.02$ |
| 2048 | $0.98 \pm 0.00$ | $0.94 \pm 0.01$ | $0.89 \pm 0.01$ | $0.87 \pm 0.03$ | $0.87 \pm 0.01$ | $0.86 \pm 0.02$ | $0.86 \pm 0.02$ | $0.84 \pm 0.02$ | $0.92 \pm 0.01$ |
| 4096 | $0.97 \pm 0.00$ | $0.94 \pm 0.01$ | $0.93 \pm 0.01$ | $0.88 \pm 0.01$ | $0.90 \pm 0.01$ | $0.88 \pm 0.02$ | $0.91 \pm 0.01$ | $0.85 \pm 0.01$ | $0.95 \pm 0.00$ |
| VAE | $0.00 \pm 0.00$ | $0.00 \pm 1.00$ | $0.02 \pm 2.00$ | $0.00 \pm 3.00$ | $0.01 \pm 4.00$ | $0.01 \pm 5.00$ | $0.02 \pm 6.00$ | $0.02 \pm 7.00$ | $0.02 \pm 8.00$ |

| | | | | PickupLightsColors | | | | DoorLights | |
| $N$ | hand_y | hand_z | obj_y | obj_z | light | table_color | obj_color | door_angle | light |
|---|---|---|---|---|---|---|---|---|---|
| 128 | $0.94 \pm 1.00$ | $0.91 \pm 2.00$ | $0.72 \pm 4.00$ | $0.31 \pm 5.00$ | $0.88 \pm 6.00$ | $0.43 \pm 7.00$ | $0.62 \pm 8.00$ | $0.89 \pm 3.00$ | $0.84 \pm 4.00$ |
| 256 | $0.95 \pm 1.00$ | $0.96 \pm 2.00$ | $0.85 \pm 4.00$ | $0.47 \pm 5.00$ | $0.95 \pm 6.00$ | $0.62 \pm 7.00$ | $0.77 \pm 8.00$ | $0.95 \pm 3.00$ | $0.92 \pm 4.00$ |
| 512 | $0.96 \pm 1.00$ | $0.97 \pm 2.00$ | $0.91 \pm 4.00$ | $0.61 \pm 5.00$ | $0.97 \pm 6.00$ | $0.79 \pm 7.00$ | $0.82 \pm 8.00$ | $0.89 \pm 3.00$ | $0.95 \pm 4.00$ |
| 1024 | $0.95 \pm 1.00$ | $0.96 \pm 2.00$ | $0.94 \pm 4.00$ | $0.69 \pm 5.00$ | $0.97 \pm 6.00$ | $0.87 \pm 7.00$ | $0.92 \pm 8.00$ | $0.91 \pm 3.00$ | $0.94 \pm 4.00$ |
| 2048 | $0.95 \pm 1.00$ | $0.98 \pm 2.00$ | $0.95 \pm 4.00$ | $0.75 \pm 5.00$ | $0.96 \pm 6.00$ | $0.90 \pm 7.00$ | $0.93 \pm 8.00$ | $0.91 \pm 3.00$ | $0.94 \pm 4.00$ |
| 4096 | $0.95 \pm 1.00$ | $0.96 \pm 2.00$ | $0.94 \pm 4.00$ | $0.80 \pm 5.00$ | $0.96 \pm 6.00$ | $0.89 \pm 7.00$ | $0.96 \pm 8.00$ | $0.92 \pm 3.00$ | $0.95 \pm 4.00$ |
| VAE | $0.08 \pm 0.00$ | $0.25 \pm 1.00$ | $0.07 \pm 2.00$ | $0.09 \pm 3.00$ | $0.24 \pm 4.00$ | $0.09 \pm 5.00$ | $0.04 \pm 6.00$ | $0.01 \pm 0.00$ | $0.34 \pm 1.00$ |

Table 4: **How many weak labels are needed to learn a sufficiently-disentangled state representation?** We trained disentangled representations on varying numbers of weakly-labelled data samples $\{(s_1^{(i)}, s_2^{(i)}, y^{(i)})\}_{i=1}^N$ ($N \in \{128, 256, \ldots, 4096\}$), then evaluated how well they disentangled the true factors of variation in the data. On the evaluation dataset, we measure the Pearson correlation between the true factor value of the input image vs. the latent dimension of the encoded image. For the VAE (obtained from SkewFit), we took the latent dimension that has the highest correlation with the true factor value. We report the 95% confidence interval over 5 seeds. Even with a small amount of weak supervision (e.g. around 1024 labels), we are able to attain a representation with good disentanglement.

## A.3 Noisy data experiments

While the weakly-labelled data can be collected at scale from crowd-sourcers and does not require expertise, the human labellers may mistakenly provide inaccurate rankings. Thus, we evaluated the robustness of the disentangled representation learning on more realistic, noisy datasets which are far less clean than the toy datasets used by Shu et al. [68].

**Real-world dataset**: We collected 1,285 RGB images (1,029 train, 256 test) on a real Franka robot with 5 blocks (see Fig. 10a). We collected the images under various indoor lighting settings and at different times of the day, but we did not provide labels for the environment lighting conditions. We found that the robot arm often caused occlusion, hiding blocks from the camera view, so we used two RGB cameras placed at different locations, and stacked the RGB images into 6 channels (i.e., image arrays of shape $48 \times 48 \times 6$). We then had a human provide weak labels for the block positions. In Fig. 10b, we show that the learned disentangled model attains a sufficiently high Pearson correlation between the true XY-position of the block (relative to the image frame) vs. the corresponding latent dimension of the encoded image. The results suggest that weakly-supervised disentangled

Figure 9: **How many weak labels are needed to help visual goal-conditioned RL?** We evaluate the performance of our method (WSC) on visual goal-conditioned tasks as we vary the number of weak pairwise labels $N \in \{128, 256, \ldots, 4096\}$. We find that 1024 pairwise labels is generally sufficient for good performance on all domains.

| Block | Pearson correlation | |
| color | x | y |
| --- | --- | --- |
| Red | $0.747 \pm 0.046$ | $0.715 \pm 0.016$ |
| Blue | $0.649 \pm 0.034$ | $0.673 \pm 0.042$ |
| Green | $0.718 \pm 0.057$ | $0.625 \pm 0.066$ |
| Yellow | $0.663 \pm 0.052$ | $0.673 \pm 0.057$ |
| Purple | $0.505 \pm 0.041$ | $0.518 \pm 0.055$ |

(a) Franka robot with 5 blocks  (b) Disentangled representation performance

Figure 10: **Real-world dataset**: *(a)*: We collected 1,285 RGB camera images (1,029 train, 256 test) on a real Franka robot with 5 block objects, and then had a human provide weak labels for the block positions. We collected the images under various lighting conditions, and used two camera viewpoints to overcome object occlusion. *(b)*: Our method attains a sufficiently high Pearson correlation between the true XY-position of the block (relative to the image frame) vs. the latent dimension of the encoded image, suggesting that weakly-supervised disentangled representation learning may be useful for training robots in the real-world. Results are taken over 6 seeds.

representation learning may be useful for training robots in the real-world, despite challenges such as environment stochasticity and object occlusion.

**Noisy labels**: We generated noisy datasets for *PushLights* with $n \in \{1, 2, 3\}$ objects, where each factor label was corrupted with probability 5% or 10%. In Table 5, we evaluate the quality of the learned disentangled representation model on the noisy datasets. Our method learns a robustly-disentangled representation with 5% noise (around 80% correlation), but achieves lower performance with 10% noise (around 60-70% correlation).

## A.4 Latent policy visualizations: WSC vs. SkewFit

We provide additional visualizations of the policy's latent space on more complex environments, previously discussed in Section 5.3. In Figure 11, we compare the latent space of policies trained by WSC and SkewFit. We see that our method produces a more semantically-meaningful goal-conditioned policy, where the latent goal values directly align with the final position of the target object. The difference between WSC and SkewFit grows larger as we increase the complexity of the environment (i.e., increase the number of objects from one to three).

## B  Algorithm implementation details

Both the disentanglement model (for WSC) and VAE (for SkewFit, RIG, HER) were pre-trained using the same dataset (size 256 or 512). A separate evaluation dataset of 512 image goals is used to evaluate the policies on visual goal-conditioned tasks. We used soft actor-critic [33] as the base RL algorithm. All results are averaged over 5 random seeds.

**PushLights $n = 1$**

| Noise | hand_x | hand_y | obj1_x | obj1_y | light | All |
|---|---|---|---|---|---|---|
| 5% | $0.952 \pm 0.012$ | $0.822 \pm 0.124$ | $0.730 \pm 0.276$ | $0.606 \pm 0.298$ | $0.875 \pm 0.094$ | $0.797 \pm 0.298$ |
| 10% | $0.721 \pm 0.507$ | $0.718 \pm 0.296$ | $0.520 \pm 0.502$ | $0.501 \pm 0.270$ | $0.730 \pm 0.279$ | $0.638 \pm 0.410$ |

**PushLights $n = 2$**

| Noise | hand_x | hand_y | obj1_x | obj1_y | obj2_x | obj2_y | light | All |
|---|---|---|---|---|---|---|---|---|
| 5% | $0.949 \pm 0.024$ | $0.793 \pm 0.226$ | $0.864 \pm 0.103$ | $0.844 \pm 0.145$ | $0.842 \pm 0.147$ | $0.873 \pm 0.032$ | $0.936 \pm 0.041$ | $0.872 \pm 0.118$ |
| 10% | $0.853 \pm 0.165$ | $0.588 \pm 0.357$ | $0.665 \pm 0.422$ | $0.518 \pm 0.506$ | $0.747 \pm 0.185$ | $0.864 \pm 0.02$ | $0.916 \pm 0.04$ | $0.736 \pm 0.400$ |

**PushLights $n = 3$**

| Noise | hand_x | hand_y | obj1_x | obj1_y | obj2_x | obj2_y | obj3_x | obj3_y | All |
|---|---|---|---|---|---|---|---|---|---|
| 5% | $0.786 \pm 0.144$ | $0.78 \pm 0.164$ | $0.728 \pm 0.217$ | $0.698 \pm 0.180$ | $0.791 \pm 0.117$ | $0.858 \pm 0.057$ | $0.877 \pm 0.013$ | $0.833 \pm 0.038$ | $0.794 \pm 0.661$ |
| 10% | $0.632 \pm 0.487$ | $0.551 \pm 0.295$ | $0.587 \pm 0.264$ | $0.547 \pm 0.315$ | $0.613 \pm 0.307$ | $0.817 \pm 0.023$ | $0.864 \pm 0.033$ | $0.851 \pm 0.068$ | $0.610 \pm 0.643$ |

Table 5: **Noisy labels**: We trained disentangled representations on noisy *PushLights* datasets for $n \in \{1, 2, 3\}$ objects, where each factor label was corrupted with probability 5% or 10%. We then measured the Pearson correlation between the true factor values vs. the corresponding latent dimension. Our method learns robustly-disentangled representations with 5% noise (around 80% correlation), but achieves lower performance with 10% noise (around 60-70% correlation). Results are taken over 5 seeds.

**Disentangled representation learning**: We describe the disentangled model network architecture in Table 7, which was slightly modified from [68] to be trained on $48 \times 48$ image observations from the Sawyer manipulation environments. The encoder is not trained jointly with the generator, and is only trained on generated data from $G(z)$ (see Eq. 1). All models were trained using Adam optimizer with $\beta_1 = 0.5$, $\beta_2 = 0.999$, learning rate 1e-3, and batch size 64 for 1e5 iterations. The learned disentangled representation is fixed during RL training (Phase 2 in Figure 3).

**Goal-conditioned RL**: The policy and Q-functions each are feedforward networks with (400, 300) hidden sizes and ReLU activation. All policies were trained using Soft Actor-Critic [33] with batch size 1024, discount factor 0.99, reward scale 1, and replay buffer size 1e5. The episodic horizon length was set to 50 for *Push* and *Pickup* environments, and 100 for *Door* environments. We used the default hyperparameters for SkewFit from [61], which uses 10 latent samples for estimating density. For WSC, we relabelled between 0.2 and 0.5 goals with $z_g \sim p(\mathcal{Z}_{\mathcal{I}})$ (see Table 6). All RL methods (WSC, SkewFit, RIG, HER) relabel 20% of goals with a future state in the trajectory. SkewFit and RIG additionally relabel 50% of goals with $z_g \sim p^{\text{skew}}(s)$ and $z_g \sim \mathcal{N}(0, I)$, respectively.

**VAE**: The VAE was pre-trained on the images from the weakly-labelled dataset for 1000 epochs, then trained on environment observations during RL training. We trained the VAE and the policy separately as was done in [61], and found that jointly training them end-to-end (i.e., with backpropagation between VAE loss and policy loss) did not perform well. We used learning rate 1e-3, KL regularization coefficient $\beta \in \{20, 30\}$, and batch size 128. The VAE network architecture and hyperparameters are summarized in Table 8.

**SkewFit+pred** (Section 5.1): We added a dense layer on top of the VAE encoder to predict the factor values, and added a MSE prediction loss to the $\beta$-VAE loss. We also tried using the last hidden layer of the VAE encoder instead of the encoder output, but found that it did not perform well.

**SkewFit+DR** (Figure 5): We tried with and without adding the VAE distance reward to the disentangled reward $R_{z_g}(s)$ in Eq. 3, and report the best $\alpha^{\text{VAE}}$ in Table 6:

$$R^{\text{DR}}(s) = R_{z_g}(s) - \alpha^{\text{VAE}} \| e^{\text{VAE}}(s) - z_g^{\text{VAE}} \| \tag{4}$$

**Computing infrastructure**: Experiments were ran on GTX 1080 Ti, Tesla P100, and Tesla K80.

| Environment | $M$ | Factors (**User-specified factor indices are bolded**) | WSC $p_{\text{goal}}$ | $\alpha^{\text{DR}}$ |
|---|---|---|---|---|
| Push $n = 1$ | 256 | hand_x, hand_y, **obj_x**, **obj_y** | 0.2 | 1 |
| Push $n = 2$ | 256 | hand_x, hand_y, **obj1_x**, **obj1_y**, obj2_x, obj2_y | 0.3 | 1 |
| Push $n = 3$ | 512 | hand_x, hand_y, **obj1_x**, **obj1_y**, obj2_x, obj2_y, obj3_x, obj3_y | 0.4 | 0 |
| PushLights $n = 1$ | 256 | hand_x, hand_y, **obj_x**, **obj_y**, light | 0.4 | 1 |
| PushLights $n = 2$ | 512 | hand_x, hand_y, **obj1_x**, **obj1_y**, obj2_x, obj2_y, light | 0.4 | 1 |
| PushLights $n = 3$ | 512 | hand_x, hand_y, **obj1_x**, **obj1_y**, obj2_x, obj2_y, obj3_x, obj3_y, light | 0.5 | 0 |
| Pickup | 512 | hand_y, hand_z, **obj_y**, **obj_z** | 0.4 | – |
| PickupLights | 512 | hand_y, hand_z, **obj_y**, **obj_z**, light | 0.3 | – |
| PickupColors | 512 | hand_y, hand_z, **obj_y**, **obj_z**, table_color, obj_color | 0.4 | – |
| PickupLightsColors | 512 | hand_y, hand_z, **obj_y**, **obj_z**, light, table_color, obj_color | 0.3 | – |
| Door | 512 | **door_angle** | 0.3 | – |
| DoorLights | 512 | **door_angle**, light | 0.5 | – |

Table 6: **Environment-specific hyperparameters**: $M$ is the number of training images. "WSC $p_{\text{goal}}$" is the percentage of relabelled goals in WSC (Alg. 1). $\alpha^{\text{DR}}$ is the VAE reward coefficient for SkewFit+DR in Eq. 4.

| Encoder $\mathcal{N}(z; \mu(s), \sigma(s))$ | Generator $G(z)$ | Discriminator Body | Discriminator $D(s_1, s_2, y)$ |
|---|---|---|---|
| **Input:**<br>$48 \times 48 \times 3$ image<br>$4 \times 4$ Conv, 32 ch, str 2<br>Spectral norm<br>LeakyReLU<br>$4 \times 4$ Conv, 32 ch, str 2<br>Spectral norm<br>LeakyReLU<br>$4 \times 4$ Conv, 64 ch, str 2<br>Spectral norm<br>LeakyReLU<br>$4 \times 4$ Conv, 64 ch, str 2<br>Spectral norm<br>LeakyReLU<br>Flatten<br>128 Dense layer<br>Spectral norm<br>LeakyReLU<br>$2 \cdot K$ Dense layer<br>**Output:**<br>$\mu, \sigma \in \mathbb{R}^K$ | **Input:**<br>$z \in \mathbb{R}^K$<br>128 Dense layer<br>Batch norm<br>ReLU<br>$3 \cdot 3 \cdot 64$ Dense layer<br>Batch norm<br>ReLU<br>Reshape $3 \times 3 \times 64$<br>$3 \times 3$ Conv, 32 ch, str 2<br>Batch norm<br>LeakyReLU<br>$3 \times 3$ Conv, 16 ch, str 2<br>Batch norm<br>LeakyReLU<br>$6 \times 6$ Conv, 3 ch, str 4<br>Batch norm<br>Sigmoid<br>**Output:**<br>$48 \times 48 \times 3$ image | **Input:**<br>$48 \times 48 \times 3$ image<br>$4 \times 4$ Conv, 32 ch, str 2<br>Spectral norm<br>LeakyReLU<br>$4 \times 4$ Conv, 32 ch, str 2<br>Spectral norm<br>LeakyReLU<br>$4 \times 4$ Conv, 64 ch, str 2<br>Spectral norm<br>LeakyReLU<br>$4 \times 4$ Conv, 64 ch, str 2<br>Spectral norm<br>LeakyReLU<br>Flatten<br>256 Dense layer<br>Spectral norm<br>LeakyReLU<br>256 Dense layer<br>Spectral norm<br>LeakyReLU<br>**Output: Hidden layer $h$** | **Input:**<br>Weakly-labelled data<br>$(s_1, s_2, y) \in \mathcal{D}$<br>$s_1$, $s_2$, $y$ $\{\pm 1\}^K$<br>Discriminator body<br>$h_1$, $h_2$ $\to (-) \to (*)$<br>Dense layer<br>Spectral norm<br>$o_1 + o_2 + o^{\text{diff}}$<br>**Output: Prediction**<br>$o_1 + o_2 + o^{\text{diff}} \in [0, 1]$ |

Table 7: **Disentangled representation model architecture**: We slightly modified the disentangled model architecture from [68] for $48 \times 48$ image observations. The discriminator body is applied separately to $s_1$ and $s_2$ to compute the unconditional logits $o_1$ and $o_2$ respectively, and the conditional logit is computed as $o^{\text{diff}} = y \cdot (h_1 - h_2)$, where $h_1, h_2$ are the hidden layers and $y \in \{\pm 1\}$.

| VAE encoder $\mathcal{N}(z; \mu(s), \sigma(s))$ | VAE decoder | Env | $\beta$ | Best latent dim $L^{\text{VAE}}$ | |
|---|---|---|---|---|---|
| | | | | WSC | SkewFit, RIG, HER |
| **Input:** $48 \times 48 \times 3$ image<br>$5 \times 5$ Conv, 16 ch, str 2<br>ReLU<br>$3 \times 3$ Conv, 32 ch, str 2<br>ReLU<br>$3 \times 3$ Conv, 64 ch, str 2<br>ReLU<br>Flatten<br>$2 \cdot L^{\text{VAE}}$ Dense layer<br>**Output:** $\mu, \sigma \in \mathbb{R}^{L^{\text{VAE}}}$ | **Input:** $z \in \mathbb{R}^{L^{\text{VAE}}}$<br>$3 \cdot 3 \cdot 64$ Dense layer<br>Reshape $3 \times 3 \times 64$<br>$3 \times 3$ Conv, 32 ch, str 2<br>ReLU<br>$3 \times 3$ Conv, 16 ch, str 2<br>ReLU<br>**Output:**<br>$48 \times 48 \times 3$ image | *Push*<br>*Pickup*<br>*Door* | 20<br>30<br>20 | 256<br>256<br>256 | 4<br>16<br>16 |

Table 8: **VAE architecture & hyperparameters**: $\beta$ is the KL regularization coefficient in the $\beta$-VAE loss. We found that a smaller VAE latent dim $L^{\text{VAE}} \in \{4, 16\}$ worked best for SkewFit, RIG, and HER (which use the VAE for both hindsight relabelling and for the actor & critic networks), but a larger dim $L^{\text{VAE}} = 256$ benefitted WSC (which only uses the VAE for the actor & critic networks).

Figure 11: **Interpretable control**: Trajectories generated by WSC (*left*) and SkewFit (*right*), where the policies are conditioned on varying latent goals $(z_1, z_2) \in \mathbb{R}^2$. For SkewFit, we varied the latent dimensions that have the highest correlation with the object's XY-position, and kept the remaining latent dimensions fixed. The blue object always starts at the center of the frame in the beginning of each episode. The white lines indicate the target object's position throughout the trajectory. For WSC, we see that the latent goal values directly align with the direction in which the policy moves the blue object.