[Reviews · NeurIPS 2020]

Review 1

Summary and Contributions: This paper uses weak supervision on goal states that involve some set of factors important to the main RL task. They show that these factors are learnable through weak supervision, that the representations reflect these factors, and that this improves downstream performance.

Strengths: Soundness: The approach is fairly intuitive w.r.t. how to encourage agents to learn useful interpretable and transferrable representations based on underlying factors of the environment. There are a few question marks that I feel need more attention in the main text, but this is more clarity / presentation than a problem with the method. Significance / novelty / relevance: I'm not aware of another work doing this, and this is significant as it's one step towards doing this in an unsupervised way. How to learn factors completely unsupervised is an open challenge, and I think it's necessary to study weakly supervised as an intermediate step. As such, I feel this work is very relevant due to it's necessary place in such a trajectory.

Weaknesses: Soundness: mostly on clarity w.r.t. how the representation learning dataset is constructed (this is only briefly discussed in the main text) as well as how the factors are chosen (this is in the appendix). These seem like very important points that need more attention to make the story stronger. I found myself searching several times for specific details on how the dataset was constructed, and it's unclear what the policy is, at least from reading in the main text. Relevance / significance: few weaknesses here, though I would argue that my above point in terms of telling a clean story hurt its delivery.

Correctness: The claims and methodology appear to be correct, though I have one question: Is the encoder objective correct? Do you mean the reconstruction error? No matter how I read that it doesn't seem correct to minimize e(z | G(z)). On the subject of the encoder: why this (shu 2020) over other unsupervised representation learners (e.g., ST-DIM from Anand 2019 or CURL from Srinivas 2020)? The former at least uses underlying factors from RAM as a means to evaluate models. On that subject, could the dataset / labels be used to *evaluate* encoders, for instance in pure supervised learning? This might be an interesting and very useful direction, particularly since you showed that this information is useful for models (though the goal conditioned learning part is crucial, as you also show).

Clarity: The story is clear, though I have concerns about presentation of information regarding the dataset used to train the representation learner and how the factors are chosen. I feel these are very important components of the story, and I spent a lot of time trying to pull out this information (e.g., from the appendix), when I think these deserve more 1st-class treatment in the paper.

Relation to Prior Work: Good overall, nice related works section.

Reproducibility: Yes

Additional Feedback: Update: I am happy with the rebuttal: the author's promise to correct and clarify w.r.t. points I brought up, and I recommend acceptance.


Review 2

Summary and Contributions: This paper proposes a framework for goal-conditioned RL with a goal representation whose structure is learned from weak human supervision. Most goal-conditioned RL methods either use the raw image as a goal, or an encoding learned with an unsupervised method such as a VAE. This paper takes as input a (relatively small) dataset of images, and asks human annotators to rank semantic attributes for pairs of image (which has higher lighting, which one has a door which is more open, etc). The algorithm operates in two phases: 1. Using the weak supervision signal from the human annotators, a disentangled representation is learning using a GAN-type loss on triplets of 2 images and one binary label. The image encoder is kept and other components discarded. 2. The image encoder is used to encode goal images into a representations, which are then used to define a reward function to train the policy. Overall I think this is a nice paper. While it is mostly a combination of existing components (goal-conditioned RL, learning disentangled representations which GANs), the resulting algorithm seems effective and could be a step towards making RL more practical as it replaces reward function design with an (arguably) easier type of human supervision, i.e. binary labeling which can be easily crowdsourced and defining semantically meaningful directions in goal space. -------- After reading the rebuttal, my recommendation of acceptance remains the same.

Strengths: Nice idea that addresses a relevant problem, the paper is very well-written, the experiments are complete with comparisons and ablations.

Weaknesses: The main weakness I see is that this still requires the user to define the meaningful questions that will be sent for crowdsourcing and which will define the goal space. I.e, the user has to decide that the position of the robot, its distance to the door, etc. may be useful information to know for downstream tasks. The authors do recognize this point in the conclusion though, and the argument that this is easier than defining reward functions for many tasks individually is reasonable.

Correctness: There are not theoretical results to check. The empirical section is well-done to my knowledge. It is nice that they include ablation experiments studying the effect of the learned distance metric for a different algorithm. There is a nice visualization showing that certain directions in the learned in goal space align with the movement of a particular object.

Clarity: Very clear and well-written.

Relation to Prior Work: I believe it's adequately discussed, although I am not completely up to date on recent work in this area.

Reproducibility: Yes

Additional Feedback:


Review 3

Summary and Contributions: The paper presents a framework for exploiting weak supervision in control. Weak supervision is used to learn disentangled representations, which are then used in a hindsight-experience-replay-like algorithm as the goal latent space.

Strengths: The paper uses weak supervision to learn disentangled representations, which are then used as goal latent space. This framework for exploiting weak supervision can potentially be extended in various ways and could be helpful for real world RL applications. The empirical evaluation is sound, the performance improvements are significant, and I really enjoyed the ablation study, which did a great job in credit assignment for different components of the algorithm. I didn't have experience in weakly supervised learning and its combination with RL, so I can't comment on the novelty of this work.

Weaknesses: 1. My major concern is the benchmark and the l_2 distance metric. In Figure 2, the relevant features are mostly (x,y) coordinates, as well as angles. In those cases, it makes sense to use l_2 as a metric for composing rewards. But I'm concerned with the versatility of WSC. What will happen if the relevant factors include something else like color? In this case, L2 does not seem to be a good choice. I think the paper could benefit from extra experiments with such relevant factors. Also, the loss in Eq 1 seems to have nothing to do with L2, so how can we ensure the learned representation is compatible with L2 distance? 2. The presentation in Sec 2 is not clear. For example, in L80, I can't understand what it means by e_I(s) = z_I. Does it mean the set {s \in S | e_I(s) = z_I}? Then what z_I is? Or is z_I the image of e_I? Then how f_I influences z_I?

Correctness: Yes

Clarity: Mostly

Relation to Prior Work: Yes

Reproducibility: Yes

Additional Feedback: I read the author response and would like to keep my score. I'm still curious if L2 distance is meaningful when color is the relevant feature.


Review 4

Summary and Contributions: A weakly supervised RL algorithm is proposed. The proposed approach learns disentangled representations and constrains the space of tasks using user-defined factors. Experiments are carried out and the proposed approach is compared with three SOTA methods, experimental results show the proposed approach is effective.

Strengths: The experimental results show the proposed approach is effective. The paper is well written.

Weaknesses: 1. The novelty of proposed approach is limited. It looks like the authors simply apply [68] to more challenging data, and there is no theoretically contribution. 2. I have some concerns for why the proposed approach works well: (1) Is the results better because the disentangled representations are learnt or the user defines the factors relevant for solving a class of tasks? (2) Does disentangled representations work well if no factors relevant for solving a class of tasks are specified by the user? (3) Does user specified factors alone works well if no disentangled representations are learnt? (4) If no factors relevant for solving a class of tasks are specified by the user, is the proposed approach infeasible? Is the time complexity on par with SOTA methods?

Correctness: The method is correct. The empirical methodology is correct.

Clarity: The paper is well written.

Relation to Prior Work: Prior work is well discussed.

Reproducibility: Yes

Additional Feedback: No further comments, please see the weakness section for comments and suggestions. ------------------ Update ------------------- The rebuttal addresses some of my concerns. I recommend accepting the paper.

[Author Response · NeurIPS 2020]

We thank the reviewers for their valuable feedback. We will incorporate all clarifications below in the final version.

**(R4) "Novelty is limited. It looks like the authors simply apply [Shu et al. 2020] to more challenging data."** The key contribution of our work is using weak supervision to accelerate RL. Shu et al. does not consider the RL problem. One challenge in achieving this goal is that we must learn representations on data much more challenging than that used in [Shu et al. 2020]. Thus, a secondary contribution of our work is a set of tricks for scaling [Shu et al. 2020] to more complex tasks (L130-147). Our experiments on twelve visual robotic manipulation tasks demonstrate how using weak supervision can help goal-conditioned RL achieve significant improvements over previous SOTA methods (Section 5.1), as well as produce interpretable policies (Section 5.3). Our ablation study (Section 5.2) also studies the relative importance of disentanglement for goal generation vs. for defining reward functions.

**(R4) "Why does the proposed approach work so well?"** The learned disentangled representation enables the RL agent to explore along meaningful axes of variation (e.g. object position), while ignoring task-irrelevant state dimensions (e.g. lighting). This results in much faster training of goal-conditioned RL (Fig 4, 5).

**(R4) "If no relevant factors are specified by the user [...] Does the disentangled representation work well? Is the proposed approach infeasible?"**

- **Is our disentangled approach better than alternative methods for utilizing the supervision?** In the main paper, we compare WSC to 'SkewFit+pred', which is a variant of SkewFit that also optimizes an auxiliary supervised prediction loss on the factor identity (see L227-233 and Fig 5). We find that SkewFit+pred performs worse than our method even though it uses stronger supervision (exact labels). This comparison suggests that disentangling meaningful and irrelevant factors is important for effectively leveraging weak supervision.

- The policy learned by WSC depends on the choice of the user-specified factor indices, $\mathcal{I}$. For example, if '$\mathcal{F}_{\mathcal{I}}$ = {hand_xy}', then the policy will learn to move the robot arm to different positions. If '$\mathcal{F}_{\mathcal{I}}$ = {blue_obj_xy, red_obj_xy}', then the policy will learn to move the blue and red objects to different positions. In many cases, it is easier and more scalable to specify the relevant axes of variation along which the agent should do exploration (e.g., "Explore by changing the object XY-position"), than to design reward functions or provide demonstrations.

**(R4) "Do user-specified factors alone work well if no disentangled representations are learnt?"** It's unclear how to use user-specified factors without learning a representation. We did run an ablation of WSC with an oracle representation that was hand-crafted to be disentangled. Experiments with this oracle representation resulted in better performance than with a learned representation. However, such oracle representations are often infeasible to acquire in the real world.

**(R4) "Is the time complexity on par with SOTA methods?"** Yes, it is the same as SkewFit and RIG.

**(R3) L2 distance metric**: Similar to findings by [SkewFit, RIG], we found that dense rewards (e.g., L2 distance in latent goal space) work better than sparse indicator rewards for training goal-conditioned RL. We will include an ablation study of WSC using different distance metrics in the final version.

**(R3) "What will happen if the relevant factors include [features] like color?"** Our method explores and learns to achieve goals that vary the relevant factors. If color is specified as a relevant factor, then the agent will attempt to change the color of its environment (e.g., by turning on colored lights or painting objects in the scene).

**(R3) Clarification about notation**: Given any vector $v \in \mathbb{R}^K$, we use $v_{\mathcal{I}} \in \mathbb{R}^{|\mathcal{I}|}$ to denote the subvector extracted from $z$ using the subindices $\mathcal{I} \subseteq \{1, \ldots, K\}$. Given an observation $s$, the encoder outputs a latent vector $z := e(s)$ in the disentangled latent space, $\mathcal{Z} \subseteq \mathbb{R}^K$. We use $Z_{\mathcal{I}} \subseteq \mathbb{R}^{|\mathcal{I}|}$ to denote the latent space restricted to the indices in $\mathcal{I}$. The true factor value of the current observation, $f_{\mathcal{I}}(s)$, is not observed by the agent, and is only used to evaluate the true goal distance: $d(f_{\mathcal{I}}(s), f_{\mathcal{I}}^*)$.

**(R1) Paper presentation**: (1) Thanks for the helpful feedback! We will move details about dataset generation (Appendix B.1) and MuJoCo environment factors (Appendix B.2) to the main text. (2) There was a typo: The encoder objective should *maximize* $e(z|G(z))$, to approximately invert the generator.

**(R1) "Why [Shu et al. 2020] over other [algorithms]?"** WSC is agnostic to the underlying representation learning algorithm. The main contribution of our work is to show how weak supervision can accelerate goal-conditioned RL. In our experiments, we chose to use [Shu et al. 2020] because it is provably guaranteed to recover the true disentangled representation under mild assumptions.

**(R1) "Could the dataset be used to evaluate encoders?"** We used a test dataset (drawn from the same distribution as the training set) to evaluate the learned representations of WSC and VAE (SkewFit) in Tables 3, 4, 5, 10b. [Shu et al. 2020] provides other useful eval metrics for disentanglement (e.g. SAP score)[1].

## Footnotes

[1] https://github.com/google-research/disentanglement_lib/tree/master/disentanglement_lib/evaluation/metrics


[Meta-Review · NeurIPS 2020]

The paper proposes a way to incorporate weak supervision, in the form of pairwise comparisons along various axes, into a goal-directed reinforcement learning framework, showing how this supervision can identify relevant latent factors for the construction of new tasks. The reviewers agree that this is a novel approach and makes an important step toward fully unsupervised approaches. As such, we are recommending acceptance.